# Non-Pharmacological Therapies for Post-Viral Syndromes, Including Long COVID: A Systematic Review

**DOI:** 10.3390/ijerph20043477

**Published:** 2023-02-16

**Authors:** Joht Singh Chandan, Kirsty R. Brown, Nikita Simms-Williams, Nasir Z. Bashir, Jenny Camaradou, Dominic Heining, Grace M. Turner, Samantha Cruz Rivera, Richard Hotham, Sonica Minhas, Krishnarajah Nirantharakumar, Manoj Sivan, Kamlesh Khunti, Devan Raindi, Steven Marwaha, Sarah E. Hughes, Christel McMullan, Tom Marshall, Melanie J. Calvert, Shamil Haroon, Olalekan Lee Aiyegbusi

**Affiliations:** 1Institute of Applied Health Research, University of Birmingham, Birmingham B15 2TT, UK; 2School of Sport, Exercise and Rehabilitation Sciences, University of Birmingham, Birmingham B15 2TT, UK; 3School of Oral and Dental Sciences, University of Bristol, Bristol BS8 1TH, UK; 4School of Health Sciences, University of East Anglia, Norwich NR4 7TJ, UK; 5Department of Microbiology, Royal Wolverhampton NHS Trust, Wolverhampton WV10 0QP, UK; 6Centre for Patient Reported Outcomes Research, Institute of Applied Health Research, University of Birmingham, Birmingham B15 2TT, UK; 7Birmingham Health Partners Centre for Regulatory Science and Innovation, University of Birmingham, Birmingham B15 2TT, UK; 8Midlands Health Data Research UK, Birmingham B15 2TT, UK; 9School of Medicine, University of Leeds, Leeds LS2 9JT, UK; 10Diabetes Research Centre, University of Leicester, Leicester LE1 7RH, UK; 11School of Dentistry, Institute of Clinical Sciences, University of Birmingham, Birmingham B5 7EG, UK; 12Institute for Mental Health, University of Birmingham, Birmingham B15 2TT, UK; 13Birmingham and Solihull Mental Health NHS Foundation Trust, Unit 1, B1, 50 Summer Hill Road, Birmingham B1 3RB, UK; 14National Institute for Health Research (NIHR), Applied Research Collaboration, Birmingham B15 2TT, UK; 15NIHR Birmingham Biomedical Research Centre, University of Birmingham, Birmingham B15 2TT, UK; 16Health Data Research UK, London WC1E 6BT, UK

**Keywords:** post-viral syndromes, PVS, COVID-19, Long COVID, post-COVID-19 condition, post-acute sequelae of SARS-CoV-2 infection (PASC), rehabilitation, systematic review, non-pharmacological intervention

## Abstract

Background: Post-viral syndromes (PVS), including Long COVID, are symptoms sustained from weeks to years following an acute viral infection. Non-pharmacological treatments for these symptoms are poorly understood. This review summarises the evidence for the effectiveness of non-pharmacological treatments for PVS. Methods: We conducted a systematic review to evaluate the effectiveness of non-pharmacological interventions for PVS, as compared to either standard care, alternative non-pharmacological therapy, or placebo. The outcomes of interest were changes in symptoms, exercise capacity, quality of life (including mental health and wellbeing), and work capability. We searched five databases (Embase, MEDLINE, PsycINFO, CINAHL, MedRxiv) for randomised controlled trials (RCTs) published between 1 January 2001 to 29 October 2021. The relevant outcome data were extracted, the study quality was appraised using the Cochrane risk-of-bias tool, and the findings were synthesised narratively. Findings: Overall, five studies of five different interventions (Pilates, music therapy, telerehabilitation, resistance exercise, neuromodulation) met the inclusion criteria. Aside from music-based intervention, all other selected interventions demonstrated some support in the management of PVS in some patients. Interpretation: In this study, we observed a lack of robust evidence evaluating the non-pharmacological treatments for PVS, including Long COVID. Considering the prevalence of prolonged symptoms following acute viral infections, there is an urgent need for clinical trials evaluating the effectiveness and cost-effectiveness of non-pharmacological treatments for patients with PVS. Registration: The study protocol was registered with PROSPERO [CRD42021282074] in October 2021 and published in BMJ Open in 2022.

## 1. Introduction

Globally, there have been over 520 million cases of COVID-19, with over 6 million associated deaths, as of 13 May 2022 [1]. The pandemic has triggered a concerted global effort to rapidly develop and deliver safe vaccinations at record speed, which have significantly reduced the morbidity, mortality, and disease transmission associated with Severe Acute Respiratory Syndrome Coronavirus-2 (SARS-CoV-2) infection [2,3,4]. Although vaccines have substantially reduced mortality, the breadth and relapsing–remitting nature of ongoing symptoms (such as fatigue and dyspnoea) that may arise from even mild infections of COVID-19 pose a substantial burden to patients and health services [5,6]. The term ‘Long COVID’, used throughout this review, is the generally preferred terminology for such ongoing symptoms following COVID-19 infection, known also as post-COVID-19 condition or post-acute sequelae of SARS-CoV-2 infection (PASC) [7,8].

According to Office for National Statistics (ONS) data, 1.7 million people living in private households in the United Kingdom (UK) (2.7% of the population) report a self-diagnosis of Long COVID, with 1.1 million (71%) having confirmed or suspected COVID-19 more than 12 weeks prior [5]. Despite the substantial prevalence of these ongoing symptoms, the current National Institute for Health and Care Excellence (NICE) guideline for the management of the Long COVID provides only limited guidance for patients, focusing on self-management, individual goal setting through shared decision making, and phased returns to work for those capable and rehabilitation [9].

Although the nature and presentations of Long COVID is still under investigation, many of its clinical features, including respiratory, neurological, psychological, and gastroenterological complications, and specifically fatigue, occur following other acute viral infections [10,11]. Such prolonged sequelae are referred to as post-viral syndromes (PVS) and follow exposure to viral pathogens, such as Epstein–Barr virus (EBV) and Chikungunya virus [12]. The effectiveness of therapies for prolonged symptoms, such as fatigue, following infection with endemic viruses, such as EBV, have previously been investigated [13,14].

A recent mixed-methods systematic review of post-viral fatigue syndromes provides useful lessons that may guide the development of Long COVID support services [14]. However, to our knowledge, beyond the synthesis of interventions used to support patients with fatigue, there is no review of the wider array of non-pharmacological interventions for patients with symptoms, such as dyspnoea or arthralgia, both common with Long COVID [6]. Therefore, there is an urgent need to identify whether the existing evidence base on similar PVS can inform the management of patients with Long COVID. Hence, this systematic review summarises the evidence on non-pharmacological treatments for PVS, including Long COVID.

## 2. Materials and Methods

This review has been conducted in accordance with the PRISMA guidelines (Appendix A) [15]. The protocol was registered on PROSPERO (CRD42021282074) in October 2021, and the full protocol has been published elsewhere [16]. For ease of reading, an overview of the research question has been described using the PICO framework below [17]:Population: The population of interest consisted of adults and children with a PVS (including Long COVID). There is no universally agreed definition of PVS, as there is heterogeneity in the temporal description of PVS onset following the initial viral exposure and additional overlap with the definition of post-viral fatigue syndrome (PVFS; ICD-10 G93.3) [18,19,20]. In line with the minimum timeframe for Long COVID used by the World Health Organization (WHO), where the temporal criteria were described, we included only those studies where post-viral symptoms lasted beyond 12 weeks [21]. However, we also included publications that provided no firm timeframe but indicated an aspect of chronicity or prolonged persistence of symptoms, as it should be noted conditions, such as PVFS, are usually denoted as a condition lasting for more than 6 months [20].Intervention and comparator: We included studies that assessed the effectiveness of non-pharmacological interventions designed to improve the symptoms of PVS against standard care, an alternative non-pharmacological therapy, or a placebo.Outcomes: The outcomes were changes in symptoms, exercise capacity, quality of life (including changes in mental and physical wellbeing) and work capability.Study type: Only randomised controlled trials (RCT) were included for patients with PVS, for conditions other than COVID-19. However, as we anticipated a lack of RCTs for SARS-CoV-2, we also included observational studies where the viral pathogen was SARS-CoV-2.

We included primary research studies published between 1 January 2001 and 29 October 2021. These dates were chosen to encompass research on the first Severe Acute Respiratory Syndrome (SARS) and Middle East Respiratory Syndrome (MERS) outbreaks, which are the two pandemic viruses most related to SARS-CoV-2 [22,23]. There were no restrictions on the setting (i.e., community or hospital based) or language.

### 2.1. Information Sources and Search Strategy

Four electronic databases were initially searched for RCTs evaluating interventions for PVS: Embase, MEDLINE, PsycINFO, and the Cumulative Index to Nursing and Allied Health Literature (CINAHL). A search strategy was developed by expanding upon keywords and combining these with Boolean operators, using the assistance of the Dudley Knowledge Library services. The search strategy used for MEDLINE is presented in Appendix A, which was adapted appropriately to search the additional databases.

As few relevant articles relating to SARS-CoV-2 were identified, we deviated from the original protocol search strategy, extending the MEDLINE search to include more terms relating to Long COVID symptoms, such as dyspnoea, fever, or breathlessness, which were noted in our previously published review as the most common symptoms associated with Long COVID [6]. These searches were then supplemented by a review of the COVID-NMA database [24], the Living Evidence on COVID-19 database [25], and the first 500 references on both MedRxiv and Google Scholar. However, we did not search for conference or symposia proceedings.

Backward and forward citation searching of the selected studies was conducted to identify further relevant studies. Where possible, the authors of protocol studies, which described a suitable trial for inclusion, were contacted to ascertain whether the findings for their trial were completed.

### 2.2. Study Selection

The study selection and quality appraisal stages of the review were facilitated using the online review software, Covidence [26]. From the search results, duplicates were removed, and then titles and abstracts were randomly allocated for screening by at least two independent reviewers. Screening criteria were defined according to whether studies: (i) included patients with a diagnosis of a PVS and (ii) documented information about non-pharmacological treatments. Any disagreements were resolved by a third independent reviewer (JSC and/or OLA).

Following the initial study screening, full-text articles were obtained and assessed according to the full inclusion and exclusion criteria. Further details on the inclusion and exclusion criteria can be found below:

Inclusion criteria:Randomised controlled trials of non-pharmacological treatments/interventions for those with PVS.Randomised controlled trials and non-randomised observational studies report non-pharmacological treatments/interventions specifically designed for Long COVID.

Exclusion criteria:Pharmacological interventions;In vitro and animal studies;Case reports/series;Systematic reviews;Non-pharmacological treatments/interventions in non-viral conditions;Protocols of trials.

At this stage, all studies were again reviewed by two independent reviewers. Any disagreements were resolved by two additional independent reviewers (JSC and/or OLA).

### 2.3. Data Extraction and Quality Appraisal

Relevant data were extracted from the included articles using a predefined extraction criterion. The data extraction was initially piloted by two reviewers (KB and NSW). The data from each study were independently extracted by at least two authors (KB, NSW, and JSC), with JSC deciding on the consensus where disparities occurred.

The extracted data included: Title, authors, country where study took place, setting, publication year, study recruitment and follow-up dates, study design, participant inclusion and exclusion criteria, sample size, baseline population characteristics for age, sex and ethnicity, virus being studied, PVS definition, reported symptoms, health outcomes reported for those with PVS, intervention description and number of patients allocated to the intervention arm, comparator description and number of patients allocated to the comparator arm, outcome of interest (including method of measurement), and description of main findings.

Included studies were appraised using the Cochrane risk-of-bias tool (RoB-2) [27], which is also the default appraisal tool in Covidence. The risk of bias was examined by at least two reviewers (KB, NSW, SCR, CM, JSC, and NB). Where there were disparities, NB adjudicated between the reviewers.

### 2.4. Narrative Synthesis

We anticipated considerable heterogeneity in the definition of PVS would limit our ability to undertake any pooled analysis or assess for publication bias. We therefore planned a narrative analysis describing the findings, characteristics, and outcomes of the included studies [28].

### 2.5. Patient and Public Involvement

The Patient and Public Involvement and Engagement (PPIE) Group for the Therapies for Long COVID Study [29] was involved in the co-development of the research question, helped determine the review’s scope and commented on the findings. The patient and public contribution was recorded using the GRIPP-2 short form checklist (Appendix A) [30].

## 3. Results

### 3.1. Description of Studies

The study selection process is outlined in Figure 1. Initially, 11,164 results were retrieved, of which 10,631 were identified as non-duplicate. Of these, 10,564 were excluded following title and abstract screening (reviewers at this stage included JSC, OLA, KB, NSK, DH, GT, SCR, RH, SHE, CM, and SH), resulting in 67 eligible studies for full-text analysis, of which 5 were suitable for inclusion. The 67 studies excluded at full-text analysis were due to either incorrect study design, incorrect patient population, or a pharmacological intervention.

Our search strategy yielded 11,164 results, which was reduced to 10,631 results after de-duplication. Following, title, and abstract screening (reviewers at this stage included JSC, OLA, KB, NSK, DH, GT, SCR, RH, SHE, CM, and SH), 67 articles were included for full text review. A total of 62 studies were then excluded (reviewers at this stage included DH, RH, KB, NSW, JSC, SH, and OLA) due to either having: (1) an incorrect study design (i.e., not a randomised controlled trial or observational study with a control group), (2) an incorrect patient population (patients did not have symptoms which were either chronic or persistent enough to fulfil our criteria for PVS), and (3) a pharmacological intervention [31,32,33,34,35,36,37,38,39,40,41,42,43,44,45,46,47,48,49,50,51,52,53,54,55,56,57,58,59,60,61,62,63,64,65,66,67,68,69,70,71,72,73,74,75,76,77,78,79,80,81,82,83,84,85,86,87,88,89]. Following the full text screening, five papers were deemed appropriate to include in the narrative synthesis (further details in Figure 1) [13,90,91,92,93].

The five studies included were RCTs, and further details of their relevant characteristics are described in Table 1 [13,90,91,92,93]. The studies were conducted in high income and upper middle-income countries, including China (*n* = 1), Norway (*n* = 1), and Brazil (*n* = 3). Adults (aged 18 years and above) formed the study cohort in four of the RCTs, with children and young people (aged 12–20 years) forming the study group for the RCT conducted in Norway. No studies reported ethnicity, one study reported an intervention relevant to patients with exposure to SARS-CoV-2, another relevant to patients experiencing EBV, and three for patients with Chikungunya virus. It is important to note that none of the studies explicitly were designed to capture the full range of symptoms experienced by the participants as part of their PVS. However, the primary symptoms captured were dyspnoea, arthralgia, fatigue (including post-exertional malaise), and general pain, which were often used as outcome markers. Additionally, as secondary outcomes, many of the studies also captured aggregated data from surveys capturing health-related quality of life.

### 3.2. Risk of Bias in Included Studies

The methodological risk of bias of the included RCTs was generally low, as seen in Figure 2. However, none of the studies blinded participants and personnel, meaning it was not possible to rule out the risk of a placebo effect or performance bias. Appendix A contains a narrative description rationalising the risk-of-bias assessment for each paper from the consensus reviewer (NB).

### 3.3. Effects of Interventions

Due to the heterogeneity in terms of viral exposure (SARS-CoV-2, EBV, or Chikungunya virus), experienced symptoms (including dyspnoea, fatigue, malaise, pain, insomnia/sleep disturbances, depression/anxiety, and arthralgia) and intervention description, the data could not be combined to perform a meta-analysis. Instead, we have narratively outlined the efficacy of the five included interventions. Table 2 summarises the intervention design, outcome measures, and key findings in each of the included trials.

### 3.4. Tele-Rehabilitation

We identified one published trial describing an intervention designed to support patients with prolonged moderate shortness of breath following exposure to SARS-CoV-2 [90]. The telerehabilitation programme in post-discharge COVID-19 patients (TERECO) was an unsupervised home-based 6-week exercise programme comprising breathing control and thoracic expansion, aerobic exercise, and lower limb muscle strength (LMS) exercise, delivered via smartphone, and remotely monitored with heart rate telemetry. Patients were recruited from three major hospitals from Jiangsu and Hubei provinces, China, and block randomised to either the intervention group or a control group consisting of those who received a 10-min standardised instruction from a physiotherapist and a written information sheet containing these instructions. Both groups were advised to maintain normal activities, adhere to a healthy diet, and follow existing public health measures (handwashing, masks, and social distancing).

In this trial, the TERECO programme was superior to the control group in the primary outcome measure at the 6-week follow-up point with regards to functional exercise capacity measured using the 6-min walking distance (6MWD) test. The mean 6MWD in the control group increased by 17.1 m (SD 63.9) from the baseline to post-treatment assessment, whereas in the TERECO group, it improved by 80.2 m (SD 74.7). The adjusted between-group difference in change in 6MWD from the baseline (treatment effect) was 65.5 m (95% CI 43.8 to 87.1; *p* < 0.001). The TERECO group maintained this benefit at 28 weeks follow-up. They also demonstrated improvements in lower limb muscle strength and the physical components of the SF-12. However, there were no improvements in spirometry-measured pulmonary function or mental health components of the SF-12. There were improvements in perceived dyspnoea measured using the modified medical research council (mMRC) score at 6 weeks post-treatment, but this improvement was not statistically significantly different at 28 weeks follow-up.

### 3.5. Music Therapy in Combination with Cognitive Behavioural Therapy

Malik et al. explored the effectiveness of a 10-week mental health training programme consisting of a combination of cognitive behavioural therapy and music therapy [13]. They recruited patients who met their eligibility criteria from the chronic fatigue following acute Epstein–Barr virus infection in adolescents (CEBA) prospective cohort study based in Norway [96]. Patients were then randomly allocated to either mental health training programme or ‘care as usual.’ Malik et al. describe that in Norway, neither general practitioners nor paediatricians schedule appointments with postinfectious chronic fatigue patients unless they have strongly reduced physical function [13]. Therefore, ‘care as usual’ implies that the relevant individuals would not receive any healthcare for their CF condition in the follow-up period. The outcomes of interest were the subjective experience of physical and mental fatigue, post-exertional malaise, pain severity, insomnia and sleep disturbances, depression and anxiety, quality of life, disability related to everyday activities, steps per day, and rates of adverse experiences. The authors found there were no statistically significant differences in any of the outcomes between the two groups. However, they noted that the study was underpowered and recommended the trial should be treated as an exploratory study as sufficient power is needed to assess the effectiveness of this treatment modality [13].

### 3.6. Resistance Exercises

Neumann et al. undertook an RCT examining the effectiveness of a resistance exercise programme for patients experiencing prolonged musculoskeletal symptoms who had been exposed to the Chikungunya virus and attended a rheumatology outpatient clinic [91]. Patients randomly allocated to the intervention arm undertook a resistance exercise programme with elastic bands over a 12-week period with a physical therapist, aiming to strengthen muscle groups that stabilise the main functional joint groups affected by Chikungunya disease. Those in the comparator group maintained their usual treatment pathway prescribed by their rheumatologist and received two weekly telephone calls to assess their symptoms. The primary outcome of interest was physical function which was assessed by the (1) 30-s chair stand test (CST), (2) 40-metre fast pace walk test (FPWT), (3) 4-step stair climb power test (4SCPT), and (4) disabilities of arm, shoulder, hand (DASH) questionnaire.

There was a significant improvement in the intervention group on the 30-s CST (*p* = 0.04, d = 0.39) at 12 weeks follow-up compared to the control group. However, there was no significant improvement in the FPWT, 4SCPT, or DASH. There was a reduction in pain intensity (secondary outcome) measured using the visual analogue scale, but there was no difference in the disease activity score measuring painful joints or SF-36 measuring quality of life. The patient’s global impression of change (PGIC) score is a validated instrument for the measurement of the perception of changing health status and treatment satisfaction in patients with chronic musculoskeletal pain and was measured only in the intervention arm, with 70% reporting improvements.

### 3.7. Pilates

De Oliveira et al. evaluated the effect of Pilates on pain reduction, improvement of joint function, and the quality of life in patients with chronic Chikungunya fever [92]. Patients were randomly allocated to either a 12-week Pilates exercise programme (two sessions per week for 50 min per session at light to moderate intensity) or a control group who did not undergo Pilates. All patients continued to receive their usual follow-up care at the Chikungunya outpatient clinic. The primary outcome was pain intensity, which was assessed using a visual analogue scale on a 0 to 10 scale on a 10 cm line. Secondary outcomes were joint range of motion, function, and quality of life. By the 12-week follow-up point those who underwent the Pilates intervention compared to the control group experienced a statistically significant improvement in pain intensity with a relative risk of a person being treated with Pilates and having decreased pain being 0.48 (95% CI 0.28–0.82), with a number needed to treat being two patients (95% CI 7–2). All secondary outcome measures showed statistically significant improvements compared to the control group.

### 3.8. Neuromodulation

Silva-Filho et al. evaluated the effects of neuromodulation in patients with Chikungunya virus on the reduction of joint pain [93]. Transcranial direct current stimulation (tDCS) is a battery-powered non-invasive neuromodulation technique in which low amplitude direct current is conducted to the cerebral cortex [93,97]. Patients were either randomised to the tDCS arm, where they experienced a constant current of 2 mA for 20 min, or the sham tDCS group, where electrodes were placed on the same position but only experienced a constant 2 mA current for 30 s. The primary outcome, pain intensity, was captured using visual analogue scales by a researcher at the baseline, after day 1 and 5 of tDCS and 1-week follow-up.

The VAS scores did not differ substantially between the active-tDCS and sham-control group at any of the four time points considered. However, in the Friedman test, a statistically significant decrease in pain over time was found only in the active-tDCS (*p* < 0.05, Friedman), and not in the sham-control group. There were also improvements in secondary outcome measures relating to pain characteristics (measured using the McGill pain questionnaire, brief pain inventory, and in the scratch flexibility test). However, there were no improvements in physical function (assessed using the hand grip test, 30-s chair stand test, 30-s arm curl test), flexibility assessed using the chair sit and reach test or health-related quality of life measured using the SF-36.

## 4. Discussion

To our knowledge, this is the first systematic review reporting the effectiveness of non-pharmacological treatments evaluated through RCTs for patients experiencing PVS including from viruses other than COVID-19 and observational studies for patients with Long COVID. We sought to synthesise knowledge that can be used to support service planning for the management of patients with Long COVID, in light of the COVID-19 pandemic. We identified five relevant trials that described five distinct types of interventions to support those experiencing chronic PVS-related symptoms. Four of the five (tele-rehabilitation, resistance exercises, Pilates, and neuromodulation) interventions reported statistically significant benefits in their primary outcomes, whereas music therapy combined with CBT did not demonstrate significant improvements in any of the measured outcomes. However, it should be noted that the study exploring music therapy with CBT was not adequately powered and further evaluation is needed to assess its efficacy. As seen in this review, the evaluation of non-pharmacological treatments for patients experiencing PVS, including Long COVID, are limited, and clinical trials are urgently needed to evaluate further therapies and confirm existing findings.

As we identified, there are limited trials of interventions designed to treat these common and persistent symptoms (fatigue being the most commonly reported in patients with Long COVID) [5,6]. Based on our study inclusion criteria, we were unable to find any trials conducted to support patients with ongoing fatigue symptoms following confirmed viral exposure. However, a recent systematic review with broader inclusion criteria, which included both pharmacological and non-pharmacological interventions to support patients with unexplained chronic fatigue syndrome/fibromyalgia, identified forty relevant trials [14]. Despite the number of these trials, the authors of that review found that relatively few approaches were effective in managing fatigue, and of those included, the existing evidence only applied to a narrow range of people with fatigue, a relatively homogeneous group of patients in an age group between 45–55 years, which is not representative of the whole patient cohort thought to experience Long COVID in the UK and elsewhere [5,14].

Although we did not find any suitable interventions to support patients with virus-related fatigue, the TERECO intervention was designed for patients following SARS-CoV-2 exposure and an mMRC score of 2–3, which indicates moderate dyspnoea, the second most commonly reported symptom of Long COVID [5,6,90]. While this study did not lead to a prolonged improvement in perceived dyspnoea, it showed improvements in the primary outcome, which was physical function [90]. Notably, the TERECO intervention is a multi-component rehabilitation intervention which, due to its remote nature, can be delivered at scale to support patients who are severely limited by their post-viral symptoms and unable to attend a clinic in person, as well as during periods of public health restrictions [90]. A recent review of rehabilitation methods delivered remotely to support physical function has demonstrated that telerehabilitation approaches could be comparable with in-person rehabilitation for a variety of chronic condition management programmes, including cardiac and pulmonary rehabilitation [98].

Although the TERECO intervention was the only published trial examining a pulmonary rehabilitation approach to support patients with Long COVID, pulmonary rehabilitation has been thoroughly evaluated in other settings, such as in patients with chronic obstructive pulmonary disease, demonstrating consistent positive outcomes in terms of dyspnoea, fatigue, and quality of life [99]. Non-controlled studies and those in populations without continuous symptoms in line with our definition of Long COVID suggest that pulmonary rehabilitation techniques are beneficial in patients who have experienced COVID-19 [32,35,40,43,45,51,52,55,58,61,64,76,77,84,100,101]. Pulmonary rehabilitation is likely to benefit those experiencing dyspnoea following COVID-19, but further trial evidence is needed to support the appropriateness, effectiveness, and cost-effectiveness in those with symptoms lasting at least 12 weeks. Such evidence might be provided by ongoing RCTs designed for this purpose, such as the Rehabilitation Exercise and Psychological Support after COVID-19 Infection (REGAIN) trial, which is evaluating the effectiveness of a rehabilitation programme for adults with ongoing COVID-19 sequelae for more than three months after hospital discharge [102].

The other interventions (Pilates, resistance exercises and neuromodulation), which we identified to be effective for patients experiencing PVS were all examined in patient populations with Chikungunya virus exposure [91,92,93]. All of these interventions identified benefits in patients’ experiences and perceptions of pain intensity, in particular arthralgia, which is another common symptom experienced by patients with Long COVID [6,91,92,93]. Although these have not yet been formally evaluated in patients with Long COVID, they may improve the symptom burden in this patient population. For example, case reports have demonstrated benefits of neuromodulation particularly in the management of the mental health effects of COVID-19 and consequently, there are RCTs underway, such as the Symptoms, Trajectory, Inequalities and Management: Understanding Long-COVID to Address and Transform Existing Integrated Care Pathways (STIMULATE-ICP) and Home-based Brain Stimulation Treatment for Post-acute Sequelae of COVID-19 (PASC) trials to examine the efficacy of this therapy modality in patients with Long COVID [103,104,105,106,107].

A key challenge with these interventions (Pilates, resistance exercises, and neuromodulation) relates to their scalability, which is particularly important when considering the scale of the public health burden in a context of limited health service capacity. However, these therapies were not assessed in trials for home use. Additionally, relating to Pilates and resistance exercises, NICE recently recommended against the use of graded exercise therapies in supporting patients with myalgic encephalitis (ME) and chronic fatigue syndrome (CFS) [108]. Although the aetiology of ME/CFS remains under investigation, patients with these conditions experience ongoing fatigue similar to Long COVID. Through expert consultation, it was deemed that people experiencing such symptoms should undertake therapy options where they remain within their energy limits and care should be given to undertake activities that do not worsen symptoms [108]. Therefore, before the widespread adoption of exercise-based therapies, such as Pilates or resistance exercises is considered for patients with Long COVID, further research is needed to identify which patients are most likely to benefit from these therapies.

Being reported in line with the PRISMA statement, our systematic review has several strengths, such as the inclusion of a comprehensive search strategy encompassing pre-print databases in addition to peer reviewed articles. The continued PPI support for this paper (as highlighted in Appendix A) was a particular strength of the paper as we were able to formulate the review design with input from people with a lived experience of Long COVID.

However, there were some limitations to the study design and some minor deviations from the published protocol. For example, the scope of the review had increased after identifying few relevant studies in the initial search, which only included patients with PVS in RCT settings. Due to the nature of our search strategy and study aim, we did not capture evidence on interventions which were examined using non-trialled and potentially less robust methods. However, we acknowledge that some such studies may indeed provide information relevant to the affected patient population and the clinicians involved in the management of Long COVID. An additional limitation we noted, during our review, was the limited nature of symptom burden reporting in the studies included. There were two challenges posed in respect to this: (1) the studies did not often describe the breadth of symptoms experienced by participants. The variety and combination of symptoms experienced by participants may impact on the effectiveness of the interventions, and (2) the studies included did not have standardised methods of reporting symptom-related outcome measures, which also limited comparability when assessing effectiveness. As such, we recommend that future research in this area is undertaken to develop standardised outcome measurement tools to facilitate the evaluation of relevant interventions pertaining to supporting patients with PVS. With the support of patients, the public, and clinicians, we have developed a symptom burden questionnaire for Long COVID (SBQ-LC) which could fulfil this need in contemporary studies [109].

## 5. Conclusions

The aim of this systematic review was to identify the existing evidence base for non-pharmacological treatments which can be delivered to support patients with PVS, including Long COVID. The key findings of this review identified few treatment/intervention modalities which have been evaluated to determine their application to patients with Long COVID. Four of the five (tele-rehabilitation, resistance exercises, Pilates, and neuromodulation) interventions reported statistically significant benefits in their primary outcomes, whereas music therapy combined with CBT did not demonstrate significant improvements in any of the measured outcomes. Considering the extensive public health burden of Long COVID, there is an urgent need for further trials to evaluate supportive interventions for chronic symptoms following SARS-CoV-2 exposure, as well as other viral pathogens, and to build upon the knowledge base across overlapping symptoms.

## Figures and Tables

**Figure 1 ijerph-20-03477-f001:**
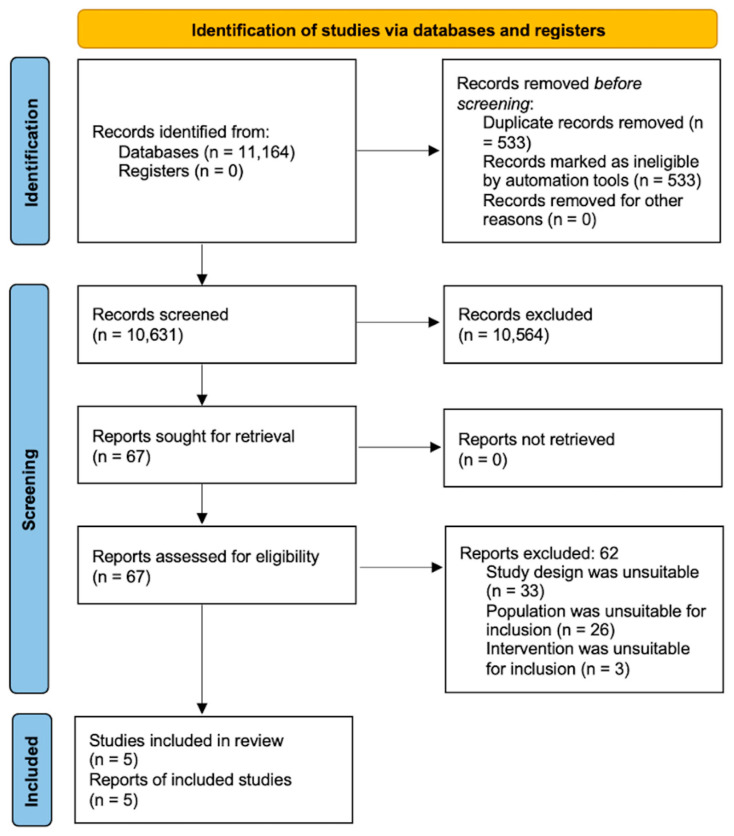
PRISMA study selection [15].

**Figure 2 ijerph-20-03477-f002:**
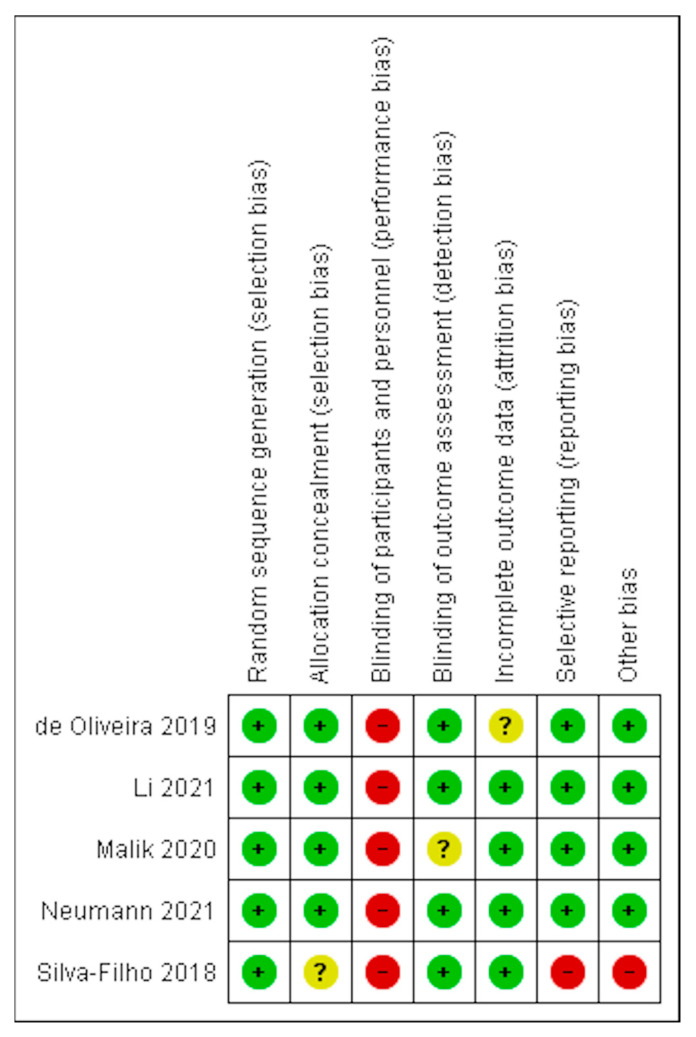
Risk of bias [13,90,91,92,93] (Legend: +: Low risk, -: High risk & ?: Unclear risk).

**Table 1 ijerph-20-03477-t001:** Study Characteristics.

Study Characteristics	Viral Definitions		Inclusion/Exclusion Criteria		Baseline Characteristics		
Authors	Virus Being Studied	Definition of Post-Viral Syndrome and Symptoms Reported by Patient Groups:	Inclusion	Exclusion	Intervention Group	Comparator Group	Overall
Authors: Li et al. [90] 2021 Country: China Setting: Hospital Study period: 2020	SARS-CoV-2	Definition: No time length specified but average time from hospital discharge to baseline was 70.07 days Symptoms: Dyspnoea	18–75 years old; Discharged from one of the three participating hospitals (three major hospitals in Jiangsu and Hubei provinces in China: Jiangsu Province Hospital/Nanjing Medical University First Affiliated Hospital, Hubei Province Hospital of Integrated Chinese and Western Medicine, and Hubei Huangshi Hospital of Chinese Medicine) after having inpatient treatment for COVID-19; and had a modified British Medical Research Council (mMRC) dyspnoea score of 2–3.	Patients with an mMRC score of 4–5 were excluded for safety reasons; Other exclusion: resting heart rate over 100 bpm, uncontrolled hypertension, uncontrolled chronic disease (e.g., diabetes with random blood glucose >16.7mmol/L, haemoglobin A1C >7.0%), cerebrovascular disease within 6 months, intra-articular drug injection or surgical treatment of lower extremities within 6 months, taking medication affecting cardiopulmonary function such as bronchodilators or beta-blockers, unable to walk independently with assistive device, unable or unwilling to collaborate with assessments, enrolled or participated in other trials within the past 3 months, having a history of severe cognitive or mental disorder or substance abuse, enrolment in other rehabilitation programme.	N = 59 Male: 27 (46%) Female: 32 (54%) Age: 49.2 (SD 10.8)	N = 60 Male: 26 (43%) Female: 34 (57%) Age: 52.0 (SD 11.1)	N = 119 Male: 53 (45%) Female: 66 (55%) Age: 50.6 (SD 11.0)
Authors: Malik et al. [13] 2020 Country: Norway Setting: Not described Study period: 2015–17	Epstein–Barr virus (EBV)	Definition: Patients had to have chronic fatigue syndrome (CF) for at least 6 months after acute infection with EBV. Symptoms: Fatigue, post-exertional malaise and pain	Developed CF 6 months after an acute EBV infection; A serological pattern indicating acute EBV infection; Age between 12 and 20 years; Living in one of the Norwegian counties Oslo, Akershus, or Buskerud	More than 6 weeks since debut of symptoms suggesting acute EBV infection; Any chronic disease that needed regular use of medication; Pregnancy.	N = 21 Male: 4 (19%) Female: 17 (81%) Age: 17.7 (SD 1.4)	N = 22 Male: 6 (27%) Female: 16 (73%) Age: 16.9 (SD 1.7)	N = 43 Male: 10 (23%) Female: 33 (77%) Age: Overall age not provided
Authors: Neumann et al. [91] 2021 Country: Brazil Setting: Hospital Study period: 2018–19	Chikungunya	Definition: Musculoskeletal symptoms lasting beyond three months Symptoms: Arthralgia	Aged 18–75 years; Serological diagnosis of Chikungunya fever and symptoms lasting 3+ months	Cognitive impairment as assessed by mini-mental state examination (MMSE); Contraindication for physical exercise (e.g., unstable angina, uncontrolled hypertension, or kidney disorder); Neurological disorders; Previous diagnosis of rheumatic disorders (except osteoarthritis); Physical impairment preventing intervention; Pregnancy; Receiving other physical modality treatments during research period; Engagement in regular physical exercise (mild- or moderate-intensity aerobic activities for 30 min five times a week or vigorous physical activity for at least 20 min three times a week)	N = 15 Male: 1 (7%) Female: 14 (93%) Age: 54.9 (SD 9.6)	N = 16 Male: 2 (12.5%) Female: 14 (87.5%) Age: 56.7 (SD 11.0)	N = 31 Male: 3 (10%) Female: 28 (90%) Age: 56.0 (SD 10.0)
Authors: Silva-Filho et al. [93] 2018 Country: Brazil Setting: Community Study period: 2016–17	Chikungunya	Definition: Positive CHIK virus for at least 6 months Symptoms: Arthralgia	Positive laboratory tests for the CHIK virus for at least 6 months (chronic phase); Preserved intellectual capacity determined by the mini mental state examination (MMSE); Physical capacity to do physical evaluation; 18 and 65 years old	Pain clearly related to any other aetiology, such as dengue, zika, rheumatoid arthritis, gout, lupus, neurologic and muscular diseases, psychiatric illness, and history of drug abuse; Signs or history of dizziness or epileptic disease; Pregnancy; Signs of severity and/or indication of hospitalization and metal implants in the head	N = 10 Male: 0 (0%) Female: 10 (100%) Age: 46.1 (SD 16.0)	N = 10 Male: 1 (10%) Female: 9 (90%) Age: 44.1 (SD 13.5)	N = 20 Male: 1 (5%) Female: 19 (95%) Age: Overall age not provided
Authors: de Oliveria et al. [92] 2019 Country: Brazil Setting: Outpatient clinical, Hospital Study period: 2017	Chikungunya	Definition: Symptoms lasting more than 3 months Symptoms: Arthralgia	18+ years; Confirmed diagnosis of Chikungunya fever; Patient in clinical treatment at Chikungunya outpatient clinic; Chronic phase of the disease (symptoms lasting more than three months)	Contraindication for physical exercise according to the treating physician; A severely limiting cognitive, auditory, visual, or motor deficit confirmed by a specialist physician; History of inflammatory, rheumatic, neurological, or neoplastic disorders	N = 22 Male: 3 (14%) Female: 19 (86%) Age: 54.4 (SD 10.6)	N = 20 Male: 0 (0%) Female: 20 (100%) Age: 59.6 (SD 9.4)	N = 42 Male: 3 (7%) Female: 39 (83%) Age: 56.9 (SD 10.6)

**Table 2 ijerph-20-03477-t002:** Intervention design, outcome measures and key findings of interventions.

Authors	Description of Intervention and Comparator	Outcome of Interest (Including Method of Measurement)	Description of Key Findings
Li et al. [90]	Intervention: Telerehabilitation programme in post-discharge COVID-19 patients (TERECO) is an unsupervised home-based 6-week exercise programme comprising breathing control and thoracic expansion, aerobic exercise, and lower limb muscle strength (LMS) exercise, delivered via smartphone, and remotely monitored with heart rate telemetry. Comparator: One-off short educational instruction at baseline	Primary outcome: - Functional exercise capacity—6-min walking test (6MWT) at 6 weeks post-treatment Secondary outcomes: - Functional exercise capacity—6 MWT at 28 weeks follow-up - Lower limb muscle strength (LMS)—Static squat test - Pulmonary function—Spirometry - Health-related quality of life (HRQOL)—Short form health survey (SF-12) - Perceived dyspnoea—modified medical research council (mMRC) score	Primary outcome: The mean 6MWD in the control group increased by 17.1 m (SD 63.9) from baseline to post-treatment assessment, whereas 6MWD in the TERECO group improved by 80.2 m (SD 74.7). The adjusted between-group difference in change in 6MWD from baseline (treatment effect) was 65.5 m (95% CI 43.8 to 87.1; *p* < 0.001). Secondary outcomes: - Functional exercise capacity—Estimated 68.6 m (95% CI 46.4 to 90.9; *p* < 0.001) treatment effect difference at follow-up. - LMS—Improved treatment effect difference 20.1s in squat position (95% CI 12.3 to 27.9; *p* < 0.001) post-treatment, and 22.2s (95% CI 14.2 to 30.2; *p* < 0.001) at follow-up. - Pulmonary function—Improvements were seen in both groups. No group differences were found apart from an improvement in maximum voluntary ventilation in the intervention group at post-treatment. - HRQOL—Statistically significant improvements in physical component score but not in mental component score. - Perceived dyspnoea—Improvements in mMRC at post-treatment but not at follow-up.
Malik et al. [13]	Intervention: The intervention consisted of a 10-week mental training programme. The patients had one introductory session followed by nine individual therapy sessions (one per week) for 1.5 h and related homework, combining elements from CBT and music therapy. Of the nine therapy sessions, four were given by a music therapist and five were given by a cognitive therapist. Comparator: Care as usual—As reported by the authors “neither General Practitioners or paediatricians in Norway schedule appointment with postinfectious CF patients unless they have strongly reduced physical function. Thus, ‘care as usual’ implies that the relevant individuals would not receive any healthcare for their CF condition in the follow-up period apart from the follow-up visits in the present study.”	Physical activity: - Steps per day—Measured used activPAL accelerometer device Symptoms: - Subjective experience of physical and mental fatigue—Chalder fatigue questionnaire [CFQ] - Post-exertional malaise—Patients were asked ‘how often do you experience more fatigue the day after an exertion?’ on a Likert scale - Pain severity—Brief pain inventory - Insomnia and other sleep disturbances—Karolinska sleep questionnaire - Depression and anxiety—Hospital anxiety and depression scale - Quality of life—Paediatric Quality of Life Inventory) - Disability related to everyday activities—Functional disability inventory - Adverse effects—Self-reported	There were no statistically significant differences between the two groups for any outcomes. Primary outcome: The mean number of steps per day decreased in the treatment group from 3 months post-baseline (7217) to 5680 at 15 months post-baseline. A decrease was also seen in the control group from 8515 at 3 months post-baseline to 7587 at 15 months. The difference between the two groups was not statistically significant: difference (95% CI) = −1298 (−4874 to 2278)).
Neumann et al. [91]	Intervention: The intervention group underwent resistance exercises with elastic bands (24 sessions over 12 weeks) supervised by a physical therapist. The exercise begins with a 5-min warm up on a stationary bike with no load, followed by resistance exercises for muscle groups that stabilize the shoulders, elbows, wrists, knees, and ankles. Comparator: Participants maintained their usual care of treatment and only received phone calls to monitor their symptoms.	Primary outcome: - Physical function: Assessed using: - 30-s chair stand test (30-s CST) - 40-m Fast-paced walk test (40 m FPWT) - 4-step Stair climb power test (4SCPT) - Disabilities of the arm, shoulder, hand (DASH) questionnaire Secondary outcomes: - Pain intensity—Visual analogue scale (VAS) and disease activity score 28 (DAS28) - Quality of life—Medical outcomes study 36-item short-form health survey (SF36) [94] - Patient’s global impression—Patient’s global impression of change (PGIC) scale (Portuguese version) only asked to intervention group	Primary outcomes: There was a significant improvement between the groups on the 30-s CST, with the resisted exercise group improving their performance compared to the control group (*p* = 0.04, d = 0.39) at 12 weeks follow-up. However, there was no significant improvement in the FPWT, 4SCPT, or DASH. Secondary outcomes: - Pain intensity—Reduction in pain intensity (*p* = 0.01; d = −0.83) but no change in painful joints count. - SF36—No significant differences. - PGIC—70% of participants in intervention group reported improvement on PGIC scale. - No adverse effects were reported in the intervention group.
Silva-Filho et al. [93]	Intervention: Patients randomised to intervention arm were given transcranial direct current stimulation (tDCS) arm where they experienced a constant current of 2 mA for 20 min. Comparator: Sham-tDCS was performed on 5 consecutive days with electrodes placed on the same position, and a constant current of 2 mA was delivered only for 30 s (10-s ramp-up) of the 20 min.	Primary outcome: - Pain intensity—Assessed using the visual analogue scale (VAS) at baseline, after day 5 of tDCS and at 1 week follow-up. Secondary outcomes: - Pain characteristics—McGill pain questionnaire and brief pain inventory ((BPI) short form) - Physical function: Assessed using: - Hand grip test—Hydraulic hand dynamometer - 30-s chair stand test - Upper limb flexion strength—30-s arm curl test - Physical flexibility: Assessed using: - Chair sit and reach test - Scratch flexibility test - Quality of life—(SF-36) [94]	Primary outcome: There was a statistically significant improvement in VAS in the tDCS group. Secondary outcomes: - Pain characteristics—There were significant improvements in both the McGill pain questionnaire and BPI int he tDCS group - Physical function—There were no statistically significant differences - Physical flexibility—No improvement in chair sit and reach test but improvement in scratch flexibility test - Quality of life—No difference in SF-36 between the groups
de Oliveira et al. [92]	Intervention: The intervention group received 24 sessions of Pilates over a 12-week period. Patients had two sessions per week for 50 min per session, and of light-to-moderate intensity (increasing the number of repetitions, starting with 6 and increasing to 12 repetitions). The exercises involved coordination, strength, flexibility, and balance. Comparator: Usual follow-up at the Chikungunya outpatient clinic, with standard clinical care for the treatment of the disease.	Primary outcome: - Pain intensity—Visual analogue scale (VAS) Secondary outcomes: - Joint range of motion—Joint goniometry - Functioning—Health assessment questionnaire - Quality of life—Short-form health survey (SF-12) [95]	Regarding the primary and secondary outcomes, in the intragroup analysis, a significant improvement was observed in all parameters after 24 Pilates sessions (week 12) in relation to the baseline (week 0), but the same was not observed in the control group. The relative risk of an individual having been treated with Pilates and having decreased pain (measured by VAS) was 0.48 (95% CI = 0.28–0.82, *p* < 0.0001) with a number needed to treat two patients (95% CI 7–2, *p* < 0.0001).

## Data Availability

The data used as part of this review can be requested from the corresponding author.

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
