# Peer review of "Non-Pharmacological Therapies for Post-Viral Syndromes, Including Long COVID: A Systematic Review"

_ijerph, 2023, doi:10.3390/ijerph20043477_

Round 1

Reviewer 1 Report

Dear Authors,

Overall the article is written well and structured. 

In my opinion, the paper has some notable changes regarding methods (Eligibility creteria and Results (Figure 1). I have provided numerous remarks and comments in the attched PDF. 

Thank you

Reviewer 2 Report

Comments on article entitled

Non-pharmacological therapies for post-viral syndromes, including Long COVID: A systematic review

In this review article, the authors have searched for trials evaluating non-pharmacological therapies for post viral syndromes (PVS). They have described and analysed five studies which met the inclusion criteria. The diseases covered are SARS CoV-2 (one study), Epstein-Barr virus (one study) and Chikungunya (three studies). Each one of the study used a different intervention for the management of PVS. Analysis of these studies has showed that the interventions, Pilates, telerehabilitation, resistance exercise and neuromodulation demonstrated some beneficial effect in management of PVS. The information presented in the review will be useful for scientists intending to study different non-pharmacological interventions for the treatment of long covid and other diseases. The manuscript is written well in a reader friendly manner. This review article is suitable for publication.

Reviewer 3 Report

Many thanks for inviting me to review this work. I enjoyed reading it and I believe it has merit. 

minor comment regarding the caption of the figures (must be added under the figure).

Conlcusion: Please highlight the most important findings which lead the reader for the recommendation. 
